# D-(+)-Galactose-induced aging: A novel experimental model of erectile dysfunction

Mathania Silva de Almeida Rezende[1], Arthur José Pontes Oliveira de Almeida[2], Tays Amanda Felisberto Gonçalves[1], Fátima de Lourdes Assunção Araújo de Azevedo[1], Sabine Helena Dantas[3], Sonaly de Lima Silva[2], Evyllen Myllena Cardoso Soares[4], Hayaly Felinto Alves[4], Thais Trajano Lima[4], Javanyr Frederico de Souza Júnior[4], Ricardo Romão Guerra[5], Islania Giselia Albuquerque Araújo[3], Isac Almeida de Medeiros[1] *

1 Programa de Pós-graduação em Produtos Naturais e Sintéticos Bioativos, Centro de Ciências da Saúde, Universidade Federal da Paraíba, João Pessoa, Paraíba, Brazil, 2 Programa de Pós-graduação em Desenvolvimento e Inovação Tecnológica em Medicamentos, Centro de Ciências da Saúde, Universidade Federal da Paraíba, João Pessoa, Paraíba, Brazil, 3 Programa de Mestrado Profissional em Gerontologia, Centro de Ciências da Saúde, Universidade Federal da Paraíba, João Pessoa, Paraíba, Brazil, 4 Centro de Ciências da Saúde, Universidade Federal da Paraíba, João Pessoa, Paraíba, Brazil, 5 Departamento de Ciências Veterinárias, Centro de Ciências Agrárias, Universidade Federal da Paraíba, Areia, Paraíba, Brazil

* isac.ufpb@gmail.com

## Abstract

Erectile dysfunction (ED) is defined as the inability to achieve and/or maintain penile erection sufficient for satisfactory sexual relations, and aging is one of the main risk factors involved. The D-(+)-Galactose aging model is a consolidated methodology for studies of cardiovascular aging; however, its potential for use with ED remain unexplored. The present study proposed to characterize a new experimental model for ED, using the D-(+)-Galactose aging model. For the experiments, the animals were randomly divided into three groups receiving: vehicle (CTL), D-galactose 150 mg/kg (DGAL), and D-(+)-galactose 150 mg/Kg + sildenafil 1.5 mg/Kg (DGAL+SD1.5) being administered daily for a period of eight weeks. All of the experimental protocols were previously approved by the Ethics Committee on the Use of Animals at the Federal University of Paraíba n˚ 9706070319. During the treatment, we analyzed physical, molecular, and physiological aspects related to the aging process and implicated in the development of ED. Our findings demonstrate for the first time that D-(+)-Galactose-induced aging represents a suitable experimental model for ED assessment. This was evidenced by an observed hyper-contractility in corpora cavernosa, significant endothelial dysfunction, increased ROS levels, an increase in cavernous tissue senescence, and the loss of essential penile erectile components.

## Introduction

Erectile dysfunction (ED) is defined as the inability to achieve and/or maintain sufficient erection for satisfactory sexual relations [1]. Its prevalence tends to increase throughout the individual's life, affecting mainly men over 40 years old [2]. With the global increase in life

**Data Availability Statement:** All relevant data are within the paper and its Supporting information files.

**Funding:** - Mathania Silva de Almeida Rezende received a scholarship from CAPES (process number 88887507825/202000) Isac Almeida de Medeiros received grant from CNPQ (process number 427783/2016-0).

**Competing interests:** The authors have declared that no competing interests exist.

expectancy, it is estimated that by the year 2025, the worldwide prevalence of ED will reach 322 million men [3].

Aging is a multifactorial process characterized by molecular, cellular, and physiological changes which increase the individual's susceptibility to the development of disease; it is also considered the main risk factor for ED [3, 4]. Of the changes observed in aging and implicated in the pathophysiology of ED we find: 1) Endothelial dysfunction, 2) Increased contractility and decreased vasodilation of the corpus cavernosum, 3) Oxidative stress, and 4) Increased vascular senescence [5, 6]. Together, these processes lead to tissue remodeling and the development of ED [6, 7].

New pharmacological tools that aim to combat and/or prevent diseases such as ED require the use of experimental animal models [8]. The accelerated D-(+)-galactose-induced aging model is widely accepted; it is based on metabolic theory, and presents many aspects similar to the natural aging process [9]. In the body, D-(+)-galactose is oxidized by galactose oxidase to form hydrogen peroxide ($H_2O_2$). In addition, this monosaccharide can react with amine groups in various proteins, forming advanced glycation products, and promoting oxidative stress [10]. Recent studies have demonstrated that chronic administration of D-(+)-galactose also leads to immune system dysregulation, sex hormone deficiencies, increasing inflammatory cytokine levels, cellular apoptosis, and diminished total antioxidant capacity [10, 11]. Taken together, these effects mimic aging and impel the subject toward development of age-related diseases [12].

However, despite D-(+)-galactose-induced accelerated aging being used as a consolidated methodology for studies of cardiovascular aging, its potential for use with ED is still unexplored. [13–15]. Thus, the present study aims to characterize a new model of ED in rats using D-(+)-galactose induced accelerated aging.

## Materials and methods

### Standards and reagents

In the present study, the following substances were used: D-(+)-galactose, Phenylephrine (Phe), Acetylcholine (ACh), sodium nitroprusside (SNP), dihydroethidium (DHE), 5-bromo-4-chloro-3-indolyl β-D-galactopyranoside (x-gal), dimethyl sulfoxide (DMSO), OCT (Optimal Cutting Temperature) Compound (Tissue Plus®), and glutaraldehyde. All were obtained from Sigma-Aldrich (Brazil). Ketamine and xylazine were purchased from Syntec (Brazil). Sildenafil was obtained from Roval Pharmacy (Brazil); heparin (Hepamax-s®) from Blau Farmacêutica S.A. (Brazil); formaldehyde 10% from Medi Química Indústria Farmacêutica Ltda (Brazil); and hematoxylin-eosin from Química Especializada Erich Ltda (Brazil). The carbogen mixture (95% $O_2$ and 5% $CO_2$) was acquired from White Martins (Brazil).

### Animals

Forty male Wistar rats (Rattus novergicus), eight weeks old, from the Animal Production Unit of the Institute for Research in Drugs and Medicines (IPeFarM) of the Federal University of Paraíba (UFPB) were used. The animals were kept under appropriate environmental conditions, temperature (22 ± 1°C), a 12-hour light-dark cycle (6–18 hours), with free access to water and food (Nuvilab CR-1, Quimtia®), while recording the physical and mental health of the animals on a daily basis. After confirmation of anesthesia induced by the intraperitoneally administration of xylazine and ketamine (10 and 75 mg/Kg, respectively), the animals were euthanized by exsanguination.

All experimental protocols were carried out according to the guidelines established by the Brazilian National Council for Animal Experiment Control (Conselho Nacional de Controle

de Experimentação Animal—CONCEA), obeying law No. 11.794/2008, submitted and previously approved by the Ethics Committee on the Use of Animals (Comissão de Ética no Uso de Animais—CEUA) of the UFPB, n° 9706070319.

## Experimental design

The animals were randomly assigned into ~~two~~ three experimental groups: the control group (CTL), which received physiological saline solution (NaCl 0.9%) intraperitoneally (IP), the D-galactose group (DGAL), which received D-(+)-galactose at 150 mg/Kg via IP, and the sildenafil group (DGAL+SD1.5) which received both D-(+)-galactose at 150 mg/Kg via IP and sildenafil 1.5 mg/Kg by oral gavage. All of the animals were subjected to eight weeks of treatment with daily administration. The IP administrations were standardized at a volume less than or equal to 2 mL/Kg [16].

The administered dose of D-(+)-galactose (150 mg/Kg) was chosen based on a review of the literature, citing doses sufficient to induce aging in the animals [17, 18]. Sildenafil was administered at 1.5 mg/Kg, corresponding (approximately) to a dose of 100 mg administered to an adult man with 70 Kg of body weight [19].

## Monitoring body weight and blood glucose

Variations in animal body weight were assessed throughout the treatment. The animals were weighed individually three times a week, always before administration of their respective treatments. The values were expressed as average weekly weight in grams (g). Glycemic analysis was performed at the end of treatment on the day of euthanasia. For this, one drop of blood was collected from the end of the caudal vein and introduced to a strip attached to an Accu-chek Guide glucometer (Roche®, Brazil). Glycemic values were expressed in mg/dL.

## Erectile function measurements—ICP/MAP ratio

Erectile function was assessed using the ICP/MAP (intra-cavernous pressure/mean arterial) pressure ratio methodology adapted according to that previously described by Kim and colleagues [20]. Briefly, at eight weeks of treatment, the animals were anesthetized with a mixture of xylazine and ketamine (10 and 75 mg/Kg, respectively, via IP). A polyethylene (PE) catheter, filled with heparinized saline (200 IU/mL) was then implanted into the right common carotid artery to the measure the mean arterial pressure (MAP).

To record intra-cavernous pressure (ICP), a 30G gauge needle, connected to a PE tube (10 mm) filled with heparinized saline (200 IU/mL), was inserted in the crural region of the left corpus cavernosum. Subsequently, the cavernous nerve was identified and a bipolar bronze stimulator (Animal Nerve Stimulating Electrode, MLA0320, ADinstruments, United States of America) was placed and electrically stimulated with 1 millisecond (ms) pulses, at 6 volts (V), and 16 Hz lasting 60 seconds (s). Two cycles of electrical stimulation were performed, the interval between each stimulation was at least 5 minutes. MAP and ICP variations were measured using pressure transducers (Disposable BP Transducer, MLT0699, ADinstruments) coupled to the PowerLab® data acquisition system (LabChart® software, version 8.1; ADInstruments, USA).

## Vascular reactivity

After euthanasia, the animal's penises were carefully isolated and immediately placed in a Krebs-Ringer nutrient solution (NaCl 118.0; KCl 4.7; CaCl₂ 2.50; KH₂PO₄ 1.20; MgSO₄ 1.17; NaHCO₃ 25.00; and glucose 5.60 (mM)) for dissection and removal of the corpus cavernosum

[21]. Corpora cavernosa were suspended vertically in isolated organ baths (Panlab Multi Chamber Organ Baths, ADIntruments, Australia) by two stainless steel metallic rods and immediately submerged in 10 mL of 37˚C Krebs-Ringer solution, with a carbogenic mixture (95% $O_2$, and 5% $CO_2$), maintained at pH 7.4, and under a stabilizing tension of 0.5 g, for 60 minutes. Voltage changes were measured using isometric transducers (MLT020, ADInstruments, Australia) and recorded in a PowerLab$^®$ data acquisition system (ML870/P, LabChart version 7.0, ADInstruments, Australia).

The contractility of the corpus cavernosum was assessed against an increasing and cumulative addition of Phe (10 nM– 300 μM), via electrical field stimulation (EFS) using different frequencies (1, 2, 4, 8, and 16 Hz) with 50 V electrical pulses of 1 ms duration. The treated groups' corpus cavernosum relaxing responses were evaluated by increasing and cumulative addition of ACh (1 nM—10 μM), and SNP (100 pM—100 μM).

## ROS measurements

Redox-sensitive fluorescent dye (DHE) was used to evaluate ROS (reactive oxygen species) formation. The corpus cavernosum was isolated and embedded in OCT compound, and then immediately frozen using liquid nitrogen for 5 minutes, before transferred and stored in a freezer at -80˚C until the next step experimentation. Microtomy of the tissue in cryostat was performed at -20˚C, in which cuts with 8 μM thickness were obtained. The tissue was fixed on slides, washed with phosphate-saline buffer (PBS) (161.0 mM NaCl; 1.8 mM $NaH_2PO_4.H_2O$, and 15.8 mM $Na_2HPO_4$), and incubated with DHE (5 μM) for 30 minutes, at 37˚C, in a humid chamber protected from light [22]. Subsequently, the sections were washed (twice) before being mounted in Fluorescence Mounting Medium (DAKO$^©$) with coverslips. Images were obtained with a Fluorescence Eclipse Ti-U Nikon$^®$ microscope (Japan). Quantification (of levels of staining) was performed using NIS-element$^®$ software. The data were normalized using the CTL group, and expressed as percentage fluorescence.

## Morphometric analysis

To perform histological sections, tissue sections of the mid-transversal part of the penis were fixed in buffered formaldehyde (10%) and incorporated into paraffin blocks with 5 μm thickness. Hematoxylin-eosin staining was used for morphometric measurement. The images were obtained using an Olympus BX-60 microscope and an Olympus camera coupled with the Olympus CellSens Dimension digital image capture program (USA). The morphometric areas were acquired using the "polygon area" function of the Olympus CellSens Dimension Program according to the given methodology, as modified by Correa et al. [23].

## Histochemical analysis of SA-β-galactosidase

Analysis of Senescence Associated β-galactosidase (SA-β-galactosidase) was adapted as previously described by Chang and colleagues [14]. The animal penile segments were embedded in OCT compound and immediately frozen in liquid nitrogen (3 min). After freezing, microtomes (5 μm) of the tissue in cryostat were performed at -20˚C. Subsequently, the tissue was washed with PBS and then fixed with a solution of formaldehyde (2%) and glutaraldehyde (0.2%), for a period of 5 minutes. In sequence, the tissues were washed with PBS and incubated with the x-gal staining solution; (150 mM NaCl, 2 mM $NaCl_2$, 5 mM $C_6N_6FeK_4$, 5 mM $C_6N_6FeK_4$, 5 mM $C_6N_6FeK_3$), 1 mg/mL of x-gal buffer, and citrate-phosphate buffer (pH 6.0 40 mM), for a maximum period of 18 h, at 37˚C, in a humid chamber protected from light [24]. Subsequently, the sections were washed with PBS solution to remove the excess x-gal

staining solution and taken immediately to analysis under a microscope (Nikon Eclipse Ti-E, Nikon, Japan).

## Statistical analysis

The data were expressed as mean ± standard error of the mean (SEM). For statistical analysis of the concentration-response curves, the maximum effect (Emax) values were used as calculated from non-linear regression of the responses obtained. The student's t-test and two-way analysis of variance (ANOVA), with the Bonferroni post-test were used. The data were considered significant when $p < 0.05$. All analyses performed were calculated using the Graph Pad Prism® version 7.0 statistical program.

## Results

### Evaluation of physical characteristics, body weights, and blood glucose levels

The animals studied presented differences in their appearance at the end of each treatment (Fig 1). The rats in the CTL group had smooth, healthy-looking, and shiny hair with uniform colors, however, the animals in the DGAL group presented curly, coarse, and opaque hair, with darker regions, and severe hair loss (Fig 1A and 1B). The animals both in the CTL and DGAL groups presented similar graduated increases in their body weights without statistical differences (n = 5; $p > 0.05$) (Fig 1C). At the end of the eight-week treatment, glycemic levels in both the CTL and DGAL animal groups (121.2 ± 4.09 mg/dL and 118.8 ± 5.73 mg/dL, respectively), were similar and without statistical differences (n = 5; $p > 0.05$).

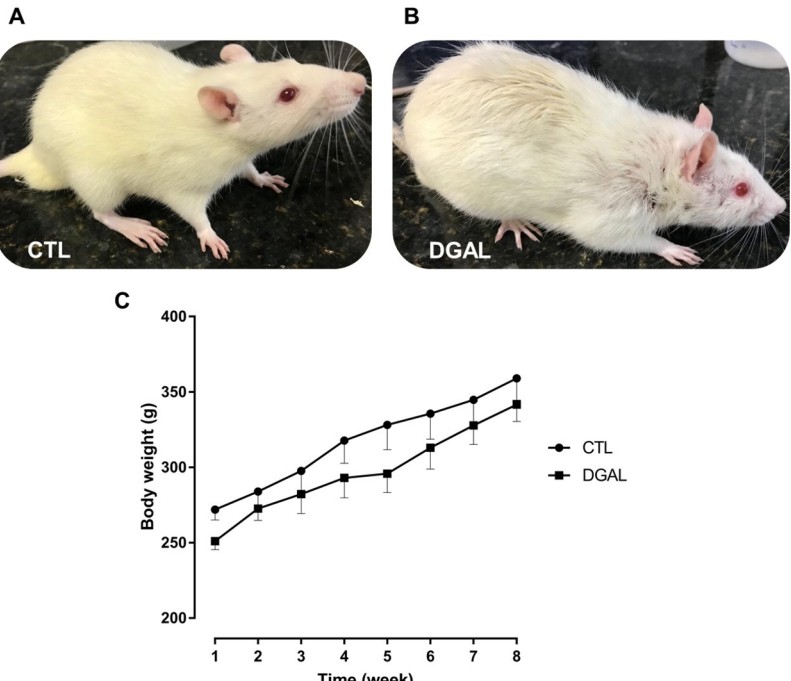

**Fig 1. Physical appearance at eight-weeks of treatment for (A) CTL, and (B) DGAL animals, (C) Average body weight (g) in function of time (8 weeks total) CTL (●) and DGAL (■).** The data are expressed as mean ± SEM (n = 5). The data were analyzed using the two-way ANOVA statistical test, followed by the Bonferroni post-test. Groups: CTL = vehicle; DGAL = D-(+)-galactose 150 mg/Kg.

## D-(+)-galactose accelerated aging model induced ED in rats

The erectile function of both groups was assessed at eight weeks of treatment. The animals in the DGAL group exhibited a significant decrease ($0.470 \pm 0.007$; n = 5; $p < 0.05$) in ICP/MAP when compared to the CTL group ($0.733 \pm 0.040$; n = 5). The animals in the DGAL+SD1.5 group showed a significant increase ($0.855 \pm 0.01$; n = 5) in ICP/MAP when compared to the DGAL group ($0.470 \pm 0.007$; n = 5; $p < 0.05$) (Fig 2).

## D-(+)-galactose accelerated aging model induced hyper-contractility and endothelial dysfunction in isolated corpus cavernosum in rats

The increasing and cumulative addition of Phe (10 nM—300 μM) for the DGAL group promoted a significant increase in contractile response (Emax = $171.95 \pm 19.24\%$; n = 5; $p < 0.05$) as compared to the CTL group (Emax = $100.00 \pm 9.44\%$; n = 5), without statistical differences in potency, according to the $pD_2$ values of the CTL groups ($pD_2 = 4.853 \pm 0.08$) and DGAL ($pD_2 = 4.93 \pm 0.09$) (Fig 3A and 3B).

The EFS (1, 2, 4, 8, and 16 Hz) in the DGAL group promoted greater contractility at all frequencies tested (1 Hz: $44.34 \pm 12.56$; 2 Hz: $74.58 \pm 16.49$; 4 Hz: $104.56 \pm 22.63$; 8 Hz: $163.13 \pm 22.23$; 16 Hz: $228.36 \pm 17.79\%$, n = 5), as compared to the CTL group (1 Hz: $13.60 \pm 6.15$; 2Hz: $21.20 \pm 9.37$; 4 Hz: $39.30 \pm 15.85$; 8 Hz: $65.50 \pm 20.56$; 16 Hz: $93.42 \pm 15.68\%$, n = 5; $p < 0.05$) (Fig 3C and 3D).

The relaxation response induced by the increasing and cumulative addition of ACh (1 nM —10 μM) was significantly lower in the DGAL group (Emax = $51.75 \pm 5.09\%$; n = 5; $p < 0.05$) when compared to the CTL group (Emax = $75.424 \pm 1.74\%$; n = 5) (Fig 3E and 3F).

The relaxation response induced by the increasing and cumulative addition of SNP (100 pM– 100 μM) did not result in a significant difference in maximum effect ($p > 0.05$). However,

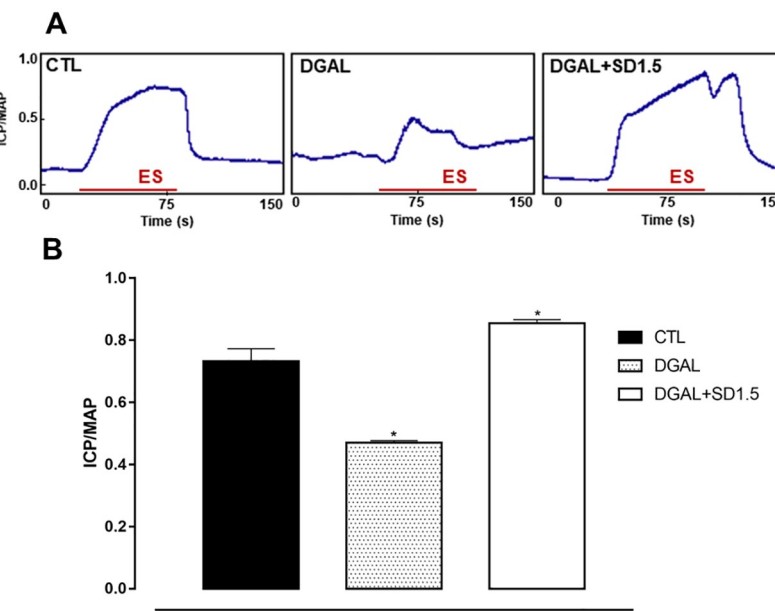

**Fig 2. Original record (A) and statistical graph (B) of the ICP/MAP in response to electrical stimulation (16 Hz, 6V, 1 ms for 60 s) of the cavernous nerve in CTL and DGAL animals at eight weeks.** Groups: CTL (vehicle); DGAL (D-(+)-galactose 150 mg/Kg); and DGAL+SD1.5 (D-(+)-galactose 150 mg/Kg + Sildenafil 1.5 mg/Kg). The results are expressed as mean ± SEM (n = 5). The data were analyzed using the Student's t-test. * $p < 0.05$ vs CTL. ES: Electrical stimulation.

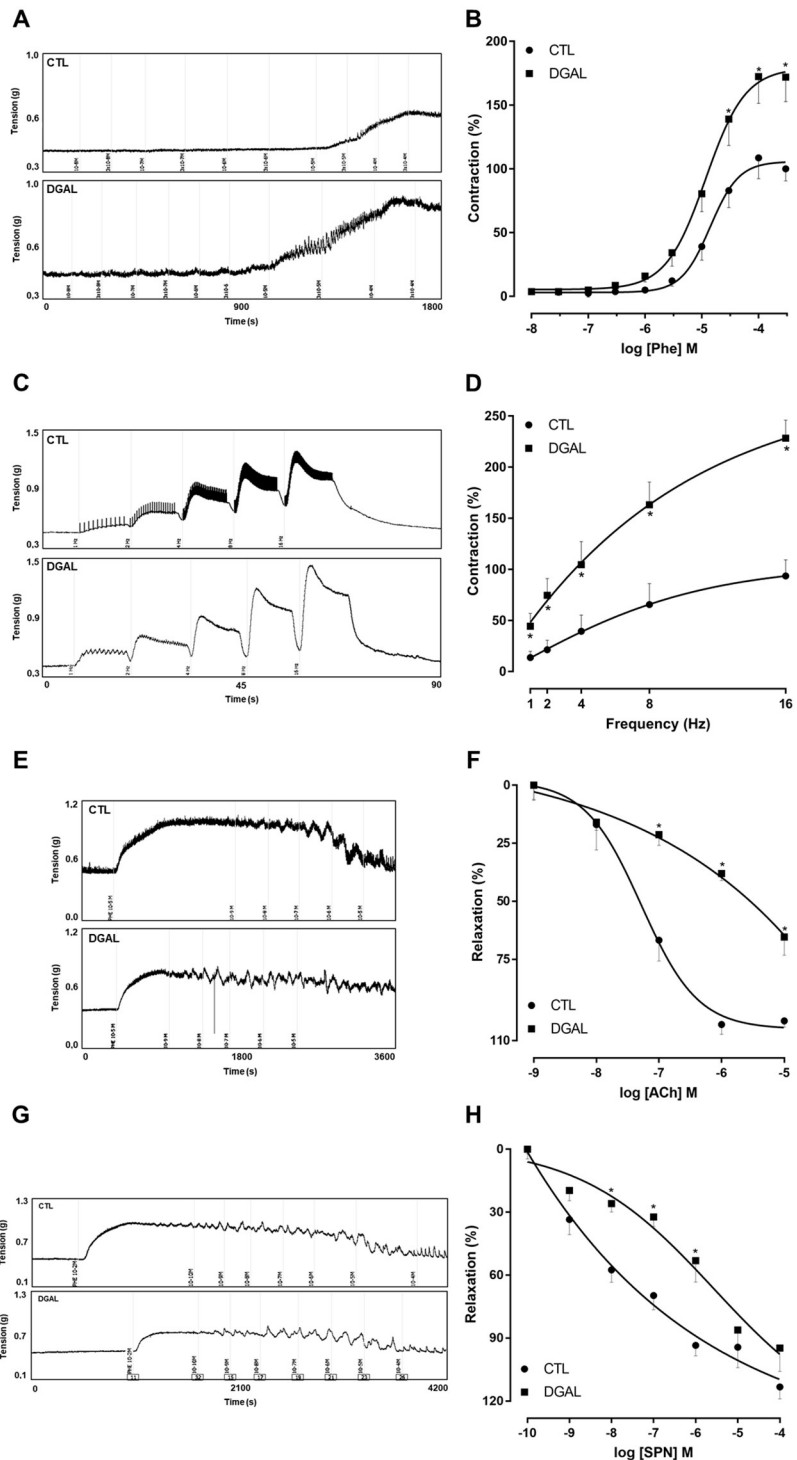

**Fig 3. Representative traces (A) and concentration-response curves (B) for Phe; representative traces (C) and frequency-response curves (D) for the electrical field stimulation EFS; representative traces (E) and concentration-response curves (F) for ACh; representative traces (G) and concentration-response curves (H) for SNP; in the corpus cavernosum isolated from rats at eight weeks of treatment, for both the CTL (●) and DGAL (■) groups.** Groups: CTL (vehicle); DGAL (D-(+)-galactose 150 mg/Kg). The results are expressed as mean ± SEM. The data were analyzed using the two-way ANOVA statistical test, followed by the Bonferroni post-test. * p < 0.05 vs CTL.

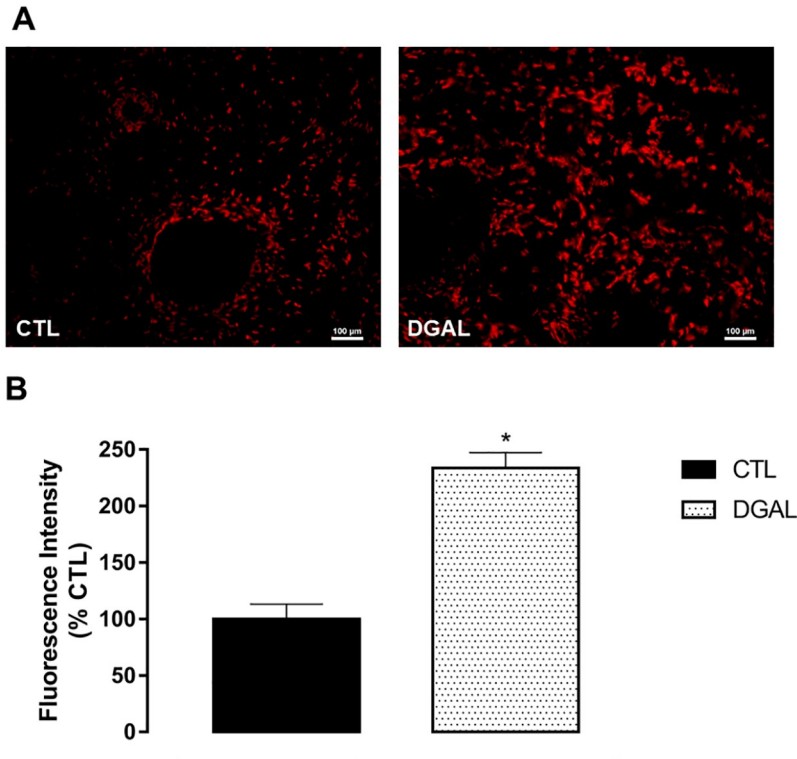

**Fig 4. Representative image (A) and quantitative analysis (B) of superoxide anion production as measured by fluorescent intensity emitted by the DHE probe.** Corpus cavernosum (8 μM) isolated from both CTL and DGAL rat groups, treated for eight weeks (20x objective). Groups: CTL (vehicle); DGAL (D-(+)-galactose 150 mg/Kg). Scale bars, 100 μm. Data are expressed as mean values of the percentage of fluorescence relative to the control ± SEM (n = 4). The data were analyzed using the Student's t-test. * $p < 0.05$ vs CTL.

there was a significant reduction in the potency ($p < 0.05$) for the DGAL group (Emax = 94.72 ± 11.04%; $pD_2$ = 6.62 ± 0.19; n = 5) as compared to the CTL group (Emax = 113.24 ± 5.59%; $pD_2$ = 7.72 ± 0.16; n = 4) (Fig 3G and 3H).

## D-(+)-galactose accelerated aging model induced increased levels of superoxide anions in the corpus cavernosum isolated from rats

Superoxide anions measurements were performed in the corpus cavernosum isolated from Wistar rats. Redox-sensitive DHE fluorescent dye was used in both the CTL and DGAL groups. The animals in the DGAL group presented a significant increase in fluorescent intensity (233.58 ± 13.69%, n = 4) when compared to the CTL group (100.00 ± 13.16, n = 4; $p < 0.05$) (Fig 4).

## D-(+)-galactose accelerated aging model induced a decrease in the total corpus cavernosum area isolated from rats

The histo-morphometry analysis of animals in the DGAL group revealed a significant decrease in the corpus cavernosum by total area (4.35x106 ± 1.83x105 μm$^2$, n = 3) when compared to the CTL group (4.99x106 ± 2.93x105 μm$^2$, n = 3; $p < 0.05$) (Fig 5).

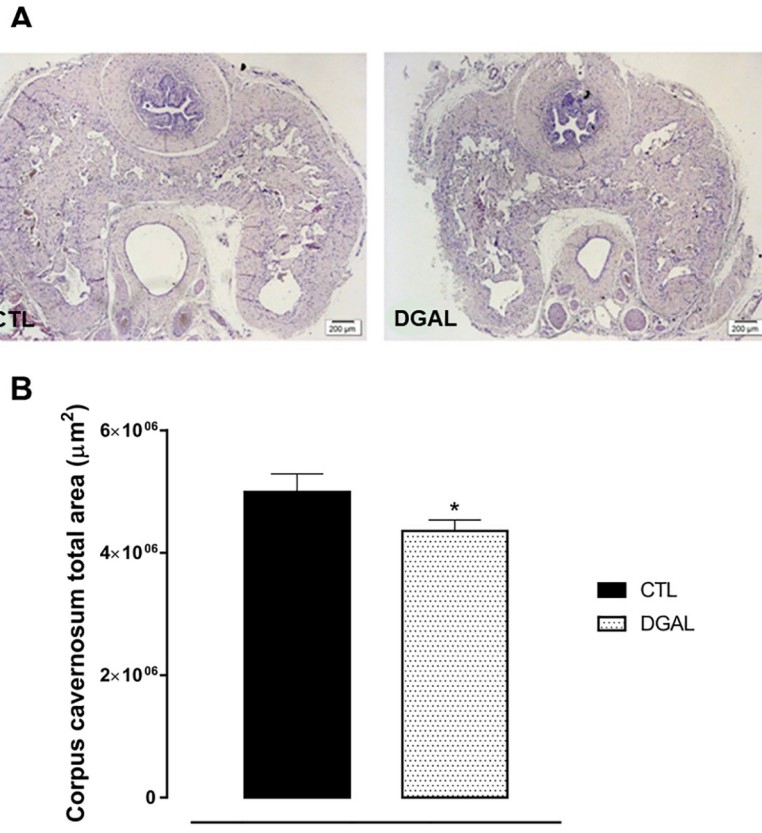

**Fig 5. Photomicrographs (A) and histomorphometric analysis (B) of the total corpus cavernosum area (8μM) isolated from rats, at eight weeks of treatment.** Groups: CTL (vehicle); DGAL (D-(+)-galactose 150 mg/Kg). Scale bars, 200 μm. The data are expressed as mean values of total corpus cavernosum area ± SEM (n = 3). The data were analyzed using the Student's t-test. * p < 0.05 vs CTL.

### D-(+)-galactose accelerated aging model induced an increase in senescence-associated β-galactosidase activity in the corpus cavernosum isolated from rats

The SA-β-galactosidase activity of animals in the DGAL group revealed a significant increase (205.189 ± 6.572%, n = 4) when compared to the CTL group (100.00 ± 13.85, n = 4; p < 0.05) (Fig 6).

## Discussion

In the present study, a novel ED model associated with mimetic aging induced by D-(+)-galactose in Wistar rats was characterized. The daily administration of 150 mg/Kg D-(+)-galactose, via IP (eight weeks), reduced erectile function *in vivo*, promoting hyper-contractility and endothelial dysfunction in isolated corpus cavernosum, as well as promoting oxidative stress, reducing the proportion of erectile components, and increasing senescence markers in penile tissue.

Chronic administration of D-(+)-galactose for a period of six to ten weeks is well described as a model to accelerate the natural aging process [25, 26]. Physiologically, the monosaccharide, is converted to glucose by galactose-1-phosphate-uridyltransferase and galactokinase [27]. Yet if in excess, deleterious metabolic disturbances are generated, with several effects

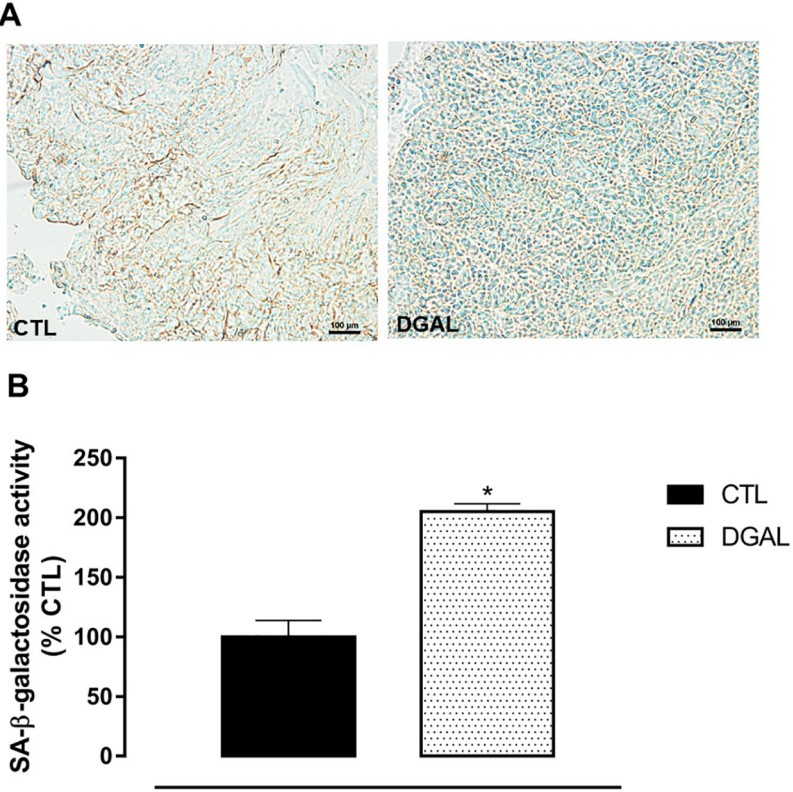

**Fig 6. Representative image (A) and quantitative analysis (B) of SA-β-galactosidase activity (%) relative to the CTL group in the corpus cavernosum (5 µM) isolated from rats, at eight weeks of treatment (20x objective).** Groups: CTL (vehicle); DGAL (D-(+)-galactose 150 mg/Kg). Scale bars, 100 µm. The data are expressed in mean percentage values (activity) in relation to the control ± SEM (n = 4). The data were analyzed using the Student's t-test. * p < 0.05 vs CTL.

such as immune system cell dysfunction, sexual hormone deficiencies, increases in inflammatory cytokines, increases in cellular apoptosis, and decreases in both total antioxidant capacity and oxidative stress (via oxidative metabolism) [26, 28, 29]. Taken together, these effects, mainly mediated by persistent oxidative stress, favor the development of disease by affecting both structure and function in pertinent tissues and organs [26, 30].

Despite D-(+)-galactose being widely used for aging research, the potential for association with ED remains unexplored. To test the hypothesis that the aging model induced by D-(+)-galactose can trigger ED, we treated Wistar rats with a chronic daily administration of D-(+)-galactose (150 mg/Kg) for eight weeks. Initially, we observed the rat's physical appearance, and at the end of the treatment period, the animals in the DGAL group presented physical characteristics such as severe hair loss, and curly or opaque hair with darker regions. This was in contrast to animals in the CTL group which presented smooth hair with a healthy look, and a bright and uniform color. Such aging characteristics were also observed in a study developed by Zhao and colleagues [29] in rats treated with D-(+)-galactose for eight weeks.

The animals' body weights were also monitored during the eight weeks of treatment. During this period it was observed that the animals of the experimental groups all similarly presented a gradual increase in their body weights, demonstrating that administration of D-(+)-galactose did not interfere in the animals' body weights. This was also observed in studies developed by Cardoso and colleagues [31]. There was also no significant change in glycemic

levels among animals in the treated groups, demonstrating that D-(+)-galactose does not inter-fere in glucose metabolism.

After the treatment period, the most used method for *in vivo* evaluation of erectile function in rats, the ICP/MAP ratio was assessed [32]. Electrical stimulation of the cavernous nerve pro-motes nitrergic discharge inducing relaxation of the corpus cavernosum with consequent ele-vation of ICP [33]. The ICP/MAP ratio in the DGAL group was reduced significantly as compared to the CTL group, demonstrating, for the first time in the literature, that the D-(+)-galactose induced aging model was effective in promoting ED. Similar results which demon-strated ED were observed in a study demonstrating that elderly rats (physiological aging) pre-sented an ICP/MAP ratio decrease [34]. Treatment with sildenafil, in animals in the DGAL +SD1.5 group, promoted a significant increase in the ICP/MAP ratio as compared to animals in the DGAL group, demonstrating that the treatment prevented ED. This result can be explained by the increase in cGMP via PDE-5 inhibition in the corpus cavernosum, as well as decreases in oxidative stress, and restoration of pro-oxidant/antioxidant equilibrium, which reduces endothelial damage and increases nitric oxide (NO) bioavailability [10, 35, 36]. These mechanisms favor relaxation of trabecular smooth muscle, and result in penile erection.

Given this *in vivo* observation of changes in erectile function, the next step would be to assess whether changes in the contractile and relaxing reactivity of corpus cavernosum isolated from the rats is involved in this process. These results are important, since erectile function is a hemodynamic process, and any imbalance is closely related to ED [37].

Therefore knowing that noradrenergic discharge and stimulation of α-adrenergic recep-tors favors increases in corpus cavernosum smooth muscle tone, and consequently impairs the state of erection [38], the response of the corpus cavernosum in contractile reactivity was evaluated using cumulative Phe and EFS curves. After the treatment period, in response to Phe and EFS, rats of the DGAL group presented increased hyper-contractility of the corpus cavernosum as compared to the CTL group. This effect may have been related to over-regu-lation of the contractile pathways in the corpus cavernosum; autonomic neuropathy (caused by exacerbation of sympathetic activity), and/or greater noradrenergic receptor sensitivity [39].

NO is another important factor and plays a key role in corpus cavernosum tonus regulation. Changes in NO synthesis or bioavailability can favor corpus cavernosum contraction, and con-sequently the development of ED [40]. We therefore evaluated whether NO release was affected by the treatments due to the action of ACh in the endothelial cells. ACh, an endothe-lial muscarinic agonist, was evaluated for its role in endothelium-dependent relaxation impairment. We observed that endothelium-dependent relaxation mediated by ACh was sig-nificantly impaired in the DGAL corpus cavernosum strips as compared to the CTL strips.

Age-related changes result in altered endothelial cell function, and cause reductions in cel-lular nitric oxide levels with subsequent impairment in penile smooth muscle relaxation. ACh, to induce its vasorelaxant effect releases NO to target muscarinic ($M_3$) receptors in endothelial cells. In our experimental conditions, animals of the DGAL group presented a significantly impaired relaxation response to ACh, as compared to the CTL group. This effect reveals an endothelial dysfunction that may be associated with decreased NO bioavailability, yielding impaired corpus cavernosum relaxation [10, 36, 39]. Lafuente-Sanchis and colleagues [41] have demonstrated that reductions in endothelium-dependent vasodilation, in response to ACh in elderly animals, is likely related to endothelial dysfunction in the cavernous trabeculae.

In addition to assessing endothelium-dependent relaxation, we also investigated impairment in pathways directly involved in relaxation of corpus cavernosum smooth muscle tissue. For this, the SNP was used, whose induced relaxation did not present statistical differ-ences between the groups in the maximum response, did promote a reduction in the potency

of the relaxation response of the DGAL group, compared to the CTL group, suggesting that the functionality of the smooth muscle cells of the corpus cavernosum may thus be altered.

Recent studies suggest that endothelial dysfunction in age-induced ED is likely related to oxidative stress [42]. Similarly, the D-(+)-galactose accelerated aging model revealed an increase in ROS levels which lead to oxidative damage [26]. Further, increased oxidative stress has also been linked to lower NO concentrations. In age-related ED, ROS has been postulated as a principal cause of impaired cavernous function. We thus evaluated whether ROS would also increase in corpus cavernosum isolated from rats using histological sections from both the CTL and DGAL groups, measuring fluorescent intensity as emitted by a DHE probe. In these experiments, the animals in the DGAL group presented a significant difference in fluorescent intensity as compared to the CTL group. This suggested an increase in superoxide anions levels, contributing to cavernous tissue remodeling, a key event in the pathophysiology of ED. Corroborating our findings, Gur and his group [43] have demonstrated an increase in ROS levels in the smooth muscle and the endothelium of the corpus cavernosum in elderly rats as compared to young animals.

In addition to functional abnormalities, age-related ED is associated with structural changes resulting in the loss of essential penile erectile components [44]. Morphologically, a significant reduction in the muscle cell layer was observed in the DGAL group, as compared to the CTL group, suggesting a loss in erectile components essential for the penile erection. This reduction functionally alters the smooth muscle of the corpus cavernosum, and revealed a significant reduction in SNP potency in the DGAL group as compared to the CTL group. Yet it is likely that such morphological changes do not sufficiently modulate functionality so as to alter the SNP response maximum. These data are in agreement with several previous studies, which reveal that both in aged men and aged animals, a decline in erectile capability is associated with a diminishing number of smooth muscle tissue cells [45–48]. Similar data were also observed in an ED model induced by diabetes [49]. Reduction of erectile function with aging has been extensively reported and related to multiple functional, morphometric, molecular, and cellular changes that lead to significant loss of erectile capability.

Accumulation of senescent cells is a biological marker of aging, and is associated with increased lysosomal SA-β-galactosidase activity. We found that in cavernous tissues, the DGAL group presented an increase in SA-β-galactosidase activity when compared to the CTL group, suggesting an accumulation of senescent cells. Similar results have been demonstrated in the cardiac tissue of animals receiving the same treatment with D-(+)-galactose [14].

D-gal is a known normal substance in the body, however, at high levels, accumulating free D-gal is converted into secondary metabolites such as galactitol, hydrogen peroxide, and Schiff's base, which in turn, induce inflammation, cellular apoptosis, and degenerative changes, this resulting in aging and age-related disorders. Further, this model was characterized by increased inflammatory cytokines, and up-regulated P16, P53, and P21 gene expression [13, 26, 50]. One of the main limitations of the present study is that the model poorly relates real physiological and biochemical changes. In addition, in the present study, inflammatory mediators, P53-P21, PI3K/Akt, and AMPK/ULK1 pathways were not measured. Nevertheless, due to its ability to mimic the senescent characteristics of natural aging, D-galactose-induced aging is potentially an ideal model for anti-aging therapeutic intervention studies.

In summary, our results demonstrate for the first time that the D-(+)-galactose aging model was able to promote ED in Wistar rats, through hyper-contractility and endothelial dysfunction in the rat corpus cavernosum. These effects may be related to oxidative stress, decreased erectile components, and accumulation of senescence cells in the corpora cavernosa of these animals.

## Conclusion

The present study reports on a novel ED rat model, successfully induced by D-(+)-galactose (daily, during 8 weeks), and validated based on functional, cellular, molecular, and morphometric analysis. The D-(+)-galactose-induced aging model was able to mimic ED in Wistar rats. The present study found in isolated rat corpus cavernosum that ED is associated with hyper-contractility and endothelial dysfunction. The effects appear to be associated with the β-galactosidase activity through an increase in oxidative stress, loss of erectile components, and increased cell senescence.

## Supporting information

**S1 File.**
(XLSX)

## Acknowledgments

The authors are grateful to Instituto UFPB de Desenvolvimento da Paraíba (IDEP) and Instituto de Pesquisa em Fármacos e Medicamentos (IPeFarM) for their technical help, respectively, for ROS measurements and Histochemical analysis of SA-β-galactosidase.

## Author Contributions

**Conceptualization:** Mathania Silva de Almeida Rezende, Arthur José Pontes Oliveira de Almeida, Sabine Helena Dantas, Isac Almeida de Medeiros.

**Data curation:** Mathania Silva de Almeida Rezende.

**Formal analysis:** Mathania Silva de Almeida Rezende, Islania Giselia Albuquerque Araújo.

**Funding acquisition:** Isac Almeida de Medeiros.

**Investigation:** Mathania Silva de Almeida Rezende, Sabine Helena Dantas.

**Methodology:** Mathania Silva de Almeida Rezende, Arthur José Pontes Oliveira de Almeida, Tays Amanda Felisberto Gonçalves, Fátima de Lourdes Assunção Araújo de Azevedo, Sabine Helena Dantas, Sonaly de Lima Silva, Evyllen Myllena Cardoso Soares, Hayaly Felinto Alves, Thais Trajano Lima, Javanyr Frederico de Souza Júnior, Ricardo Romão Guerra.

**Project administration:** Mathania Silva de Almeida Rezende, Isac Almeida de Medeiros.

**Resources:** Isac Almeida de Medeiros.

**Supervision:** Islania Giselia Albuquerque Araújo, Isac Almeida de Medeiros.

**Validation:** Mathania Silva de Almeida Rezende, Islania Giselia Albuquerque Araújo, Isac Almeida de Medeiros.

**Visualization:** Mathania Silva de Almeida Rezende, Arthur José Pontes Oliveira de Almeida, Isac Almeida de Medeiros.

**Writing – original draft:** Mathania Silva de Almeida Rezende, Arthur José Pontes Oliveira de Almeida, Tays Amanda Felisberto Gonçalves, Islania Giselia Albuquerque Araújo.

**Writing – review & editing:** Mathania Silva de Almeida Rezende, Isac Almeida de Medeiros.

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
