## [Decision Letter · Decision Letter 0]

23 Nov 2020

PONE-D-20-33703

D-[+]-Galactose-induced aging:  a novel experimental model of erectile dysfunction

PLOS ONE

Dear Dr. Medeiros,

Thank you for submitting your manuscript to PLOS ONE. After careful consideration, we feel that it has merit but does not fully meet PLOS ONE’s publication criteria as it currently stands. Therefore, we invite you to submit a revised version of the manuscript that addresses the points raised during the review process.

We look forward to receiving your revised manuscript.

Kind regards,

Michael Bader

Academic Editor

PLOS ONE

Journal Requirements:

2. Please note that PLOS does not permit references to “data not shown.” Authors should provide the relevant data within the manuscript, the Supporting Information files, or in a public repository. If the data are not a core part of the research study being presented, we ask that authors remove any references to these data.

3. Please ensure you have discussed any potential limitations of your study in the Discussion.

4.At this time, we request that you  please report additional details in your Methods section regarding animal care, as per our editorial guidelines:

(1) Please state the number of rats used in the study

(2) Please include the method of euthanasia. Please also clarify whether animals were euthanised prior to isolation of the penis.

(7) Please describe the care received by the animals, including the frequency of monitoring and the criteria used to assess animal health and well-being.

Thank you for your attention to these requests.

5. Please ensure you have correctly described the number of experimental groups in line 92.

6. At this time, we ask that you please provide scale bars on the microscopy images presented in Figures 4 and 6, and refer to the scale bar in the corresponding Figure legend.

7.We note that you have indicated that data from this study are available upon request. PLOS only allows data to be available upon request if there are legal or ethical restrictions on sharing data publicly. For more information on unacceptable data access restrictions, please see http://journals.plos.org/plosone/s/data-availability#loc-unacceptable-data-access-restrictions.

Reviewers' comments:

Reviewer's Responses to Questions

**Comments to the Author**

1. Is the manuscript technically sound, and do the data support the conclusions?

Reviewer #1: Yes

Reviewer #2: Yes

2. Has the statistical analysis been performed appropriately and rigorously? 

Reviewer #1: Yes

Reviewer #2: Yes

3. Have the authors made all data underlying the findings in their manuscript fully available?

Reviewer #1: Yes

Reviewer #2: Yes

4. Is the manuscript presented in an intelligible fashion and written in standard English?

Reviewer #1: No

Reviewer #2: Yes

5. Review Comments to the Author

Reviewer #1: The Authors have conducted a new experimental model of DE, using the D-(+)-Galactose aging model. The Authors found that the D-(+)-Galactose-induced aging represents a new experimental model suitable for the assessment of ED, evidenced by the hypercontractility of the corpora cavernosa, plus a significant endothelial dysfunction.

The logic and presentation of the key idea is not bad. I have following concerns about this work at least in its present form.

1. There was a significant difference in body weight between the CTL group and the DGAL group.

2. There are some typing, writing and grammatical errors in this manuscript. This work does require an extensive edit by a native-English speaker.

Reviewer #2: I thank the researchers for their valuable work and meticulously prepared manuscript. The research is aimed at creating a new animal model of erectile dysfunction. I believe that the comments I have stated in the attachment will increase the power of the publication.

6. PLOS authors have the option to publish the peer review history of their article (what does this mean?). If published, this will include your full peer review and any attached files.

Reviewer #1: No

Reviewer #2: **Yes: **Guldem Mercanoglu

---

## [Author Response · Author response to Decision Letter 0]

6 Feb 2021

Response to the editor

Answer: The PLOS ONE’s style requirements have been met.

2. Please note that PLOS does not permit references to “data not shown.” Authors should provide the relevant data within the manuscript, the Supporting Information files, or in a public repository. If the data are not a core part of the research study being presented, we ask that authors remove any references to these data.

Answer: References to "data not shown" have been removed. However, we have performed a new set of experiments to follow reviewer’s suggestions, where we have constructed concentration-response curves to SNP in DGAL treated rats. These new data were presented and discussed in the revised the version of manuscript.

3. Please ensure you have discussed any potential limitations of your study in the Discussion.

Answer: As we know, D-gal is a normal substance in the body, however, at high levels, accumulating free D-gal is converted into secondary metabolites such as galactitol, hydrogen peroxide, Schiff’s base, that, in turn, induce inflammation, cellular apoptosis, degenerative changes, resulting in aging and age-related disorders. Furthermore, this model is characterized by increased inflammatory cytokines, up-regulated P16, P53 and P21 genes expressions [13, 26, 50]. So that one of the main limitations of the present study is that this model can be hard to reflect real physiology and biochemistry changes. In addition, we have not measured inflammatory mediators, P53-P21, PI3K/Akt and AMPK/ ULK1 pathways in the present study. Regardless, D-galactose- induced aging carry the potential to be an ideal model for anti-aging therapeutic interventions studies due to its abilities to mimic senescence characteristics of natural aging. This paragraph has been added to the discussion section (see sentences 438-449).

4. At this time, we request that you please report additional details in your Methods section regarding animal care, as per our editorial guidelines:

(1) Please state the number of rats used in the study

Answer: We added the information concerning the number of animals used in the study (see sentence 85).

“Forty male Wistar rats (Rattus novergicus), eight weeks old, from the Animal Production Unit of the Institute for Research in Drugs and Medicines (IPeFarM) of the Federal University of Paraíba (UFPB) were used.”

(2) Please include the method of euthanasia. Please also clarify whether animals were euthanised prior to isolation of the penis.

Answer: The euthanasia method was added to the manuscript (sentence 90-93), as well as if the animals were euthanized before the isolation of the corpus cavernosum (sentence 141).

Sentence 90-93: “The animals were euthanized by exsanguination, after confirmation of anesthesia induced by the intraperitoneally administration of xylazine and ketamine (10 and 75 mg/Kg, respectively).”

Sentence 141: “After euthanasia, the animal's penises was carefully isolated and immediately placed in a Krebs-Ringer nutrient solution with composition (mM): NaCl 118.0; KCl 4.7; CaCl2 2.50; KH2PO4 1.20; MgSO4 1.17; NaHCO3 25.00 and glucose 5.60, for dissection and removal of the corpus cavernosum (21).”

(3) Please describe the care received by the animals, including the frequency of monitoring and the criteria used to assess animal health and well-being.

Answer: Animal care was described, as well as the frequency of monitoring and the criteria used to assess the animal's health and well-being (see sentence 87-90 and 94-98).

Sentence 87-90: “The animals were kept under appropriate environmental conditions, temperature (22 ± 1 ° C), 12-hour light-dark cycle (6-18 hours), with free access to water and food (Nuvilab CR-1, Quimtia®), keeping track of the physical and mental health of the animals on a daily basis.”

Sentence 94-98: “All experimental protocols were carried out according to the guidelines established by the brazilian National Council for Animal Experiment Control (Conselho Nacional de Controle de Experimentação Animal - CONCEA), obeying the law 11.794/2008, submitted and previously approved by the Ethics Committee on the Use of Animals (Comissão de Ética no Uso de Animais - CEUA) of the UFPB, nº 9706070319.”

5. Please ensure you have correctly described the number of experimental groups in line 92.

Answer: We appreciate the suggestions. Typing errors were corrected in the manuscript (see sentence 100).

“The animals were randomly assigned into three experimental groups: control group (CTL), which received physiological saline solution (NaCl 0.9%) intraperitoneally (IP), D-galactose group (DGAL), which received D-(+)-galactose 150 mg/Kg via IP and sildenafil group (DGAL+SD1.5) which received D-(+)-galactose 150 mg/Kg via IP and sildenafil 1.5 mg/Kg by oral gavage.”

6. At this time, we ask that you please provide scale bars on the microscopy images presented in Figures 4 and 6, and refer to the scale bar in the corresponding Figure legend.

Answer: The scale bars of the images were added, as requested, and the reference to it in the caption of the corresponding figure (see figure 4 and 6) (see sentence 278, 289 and 302).

Sentence 278: Scale bars, 100 μm.

Sentence 289: Scale bars, 200 μm.

Sentence 302: Scale bars, 100 μm.

7. We note that you have indicated that data from this study are available upon request. PLOS only allows data to be available upon request if there are legal or ethical restrictions on sharing data publicly. For more information on unacceptable data access restrictions, please see http://journals.plos.org/plosone/s/data-availability#loc-unacceptable-data-access-restrictions.

Answer: We have mentioned (see sentences 94-98) that all experimental protocols were carried out according to the guidelines established by the brazilian National Council for Animal Experiment Control (Conselho Nacional de Controle de Experimentação Animal - CONCEA), obeying the law 11.794/2008, submitted and previously approved by the Ethics Committee on the Use of Animals (Comissão de Ética no Uso de Animais - CEUA) of the Federal University of Paraiba-UFPB, nº 9706070319. 

Importantly, none of the funding institutions restricts the availability of data. All data will be fully available. A file containing all results spreadsheets, was submitted along with manuscript.

Response to Reviewers

1. There are minor typing errors:

Line 28- DE replace to ED

Line 348-the activity has been replaced by The activity

Answer: We thank the reviewers for the thoughtful suggestions and for the good insights for improving our manuscript. Here we tried to answer in the best form possible to all the concerns. Certainly, all the criticisms helped us to improve our manuscript and make it clearer.

The typing errors were corrected in the manuscript.

Sentence 28: Thus, the present study proposed to characterize a new experimental model of ED, using the D-(+)-Galactose aging model.

Sentence 376: The activity of ACh, which is an endothelial muscarinic agonist, was evaluated to determine if there is any impairment of endothelium-dependent relaxation mediated by this molecule.

2. In the material-method section, it should be stated how the D-(+)-galactose and sildenafil doses are selected. Because, as it is known, the dose of the research molecule is the main parameter in studies investigating molecular mechanisms.

Answer: We appreciate the pertinent suggestions. The choice of doses of both, D-galactose and sildenafil, was duly justified in the manuscript (see sentence 108-111).

The administered dose of D-(+)-galactose (150 mg/Kg) was chosen based on the literature review, in sufficient doses to induce aging in the animal (17, 18). Sildenafil was administered in a dose of 1.5 mg/Kg, corresponding, approximately, to a dose of 100 mg administered to an adult man with 70 Kg of body weight, according to the literature (19).

3. In the result part, why were DGAL + SD1.5 results not presented in the relaxation study results (between the 232-235th sentences)?

Answer: This manuscript presents a new model of erectile dysfunction induced by administration of d-galactose however, we have not investigated the in vitro effects of sildenafil in this model. Sildenafil was used only as a pharmacological tool just to check (in vivo) the effectiveness of the model.

4. In the discussion section, the reason for the improvement in ICP / MAP in the DGAL + SD1.5 group was discussed as cGMP increase mediated by PDE-5 inhibition by sildenafil (324-329th sentences). However, recent studies have shown that sildenafil increases NO bioavailability by reducing oxidative stress and restoring pro-oxidant / antioxidant balance (Leal MAS et al. Sildenafil reduces aortic endothelial dysfunction and structural damage in spontaneously hypertensive rats: Role of NO, NADPH and COX-1 pathways. Vascul Pharmacol 124:106601; 2020). Considering the possible mechanism of D-galactose in aging "oxidation to form hydrogen peroxide by galactose oxidase, ROS increase resulting in decreased SOD level and impaired redox homeostasis", (Azman KF, Zakaria R. D-galactose-induced accelerated aging model: an overview. Biogerontology 20:763-782, 2019). I believe that discussing the effect of sildenafil mentioned above in this section will strengthen the results of the study.

Answer: We rewrite this topic to follow the reviewer suggestion. We added, in the discussion section, the mechanisms that may be involved in the response of sildenafil to improve the ICP/MAP parameter (see sentence 353-356).

“This result can be explained by the increase in cGMP by inhibition of PDE-5 in the corpus cavernosum, as well as by the decrease in oxidative stress, restoration of the pro-oxidant/antioxidant balance that reduces endothelial damage and increases the bioavailability of NO (10, 35, 36). These mechanisms favor the relaxation of the smooth muscles of the trabeculae resulting in penile erection.”

5. In the discussion section, it was not understood what was meant by SHR-CTL and WKY-CTL groups between sentences 344-353. In the material method part, no information about these groups is given.

Answer: Typing errors were corrected throughout the manuscript. For instance, we replaced WKY-CTL with CTL and SHR-CTL with DGAL (see sentence 380-381).

“Thus, in the present study, endothelium-dependent relaxation mediated by ACh was significantly impaired in the strips of the corpora cavernosa of the DGAL when compared to the CTL group.”

6. In the discussion section, between the 354-363th sentences Could the decrease in acetylcholine response be due to a decrease in NO bioavailability rather than a decrease in NO release in DGAL group? The results in the DGAL + SD1.5 group specified in Article 3 can be discussed considering. Because as it is known, in the presence of ROS, ONOO- produced by interacting with NO and ROS. This molecule is the main molecule responsible for nitrosative damage and reduced NO bioavailability. Again, as discussed in Article 4; it was shown that, Sildenafil reverses endothelial dysfunction in spontaneous hypertensive rats by improving vascular relaxation to acetylcholine with increased NO bioavailability via reducing oxidative stress (Azman KF, Zakaria R. D-galactose-induced accelerated aging model: an overview. Biogerontology 20:763-782, 2019).

Answer: We appreciate the suggestion and we have introduced some information on the topic, that included a decrease in the bioavailability of NO in response to the ACh of the DGAL group (see sentence 387-389).

“This effect revealed an endothelial dysfunction that may be associated with a decrease in the bioavailability of NO, with consequent impaired relaxation of the corpus cavernosum (10, 36, 39).”

7. In the discussion section, in the paragraph starting with the 385th sentence, morphological changes are mentioned, and a decrease in smooth muscle cells in the DGAL group is stated. Does this not contradict the nitroprusside non-response referred to in sentences 364-369? Researchers should discuss this and, if necessary, state that the morphological changes are not at a level to create functional changes.

Answer: Answer: Good point. Following the reviewer suggestion, we decided to explore this data in depth in the manuscript. Concentration-response curves to SNP in DGAL treated rats were added to the manuscript. 

Fig.(see Figure in the file named Response to Reviewers) Concentration-response curves to SNP in the corpus cavernosum isolated from rats after eight weeks treatment, of both CTL (●) and DGAL (■) groups.

We found that the relaxing response induced by increasing and cumulative addition of SPN (100 pM – 100 µM) did not show significative difference in the maximum effect (p > 0.05), however, there was a significant reduction in the potency (p < 0.05) of the DGAL group (Emax = 94.72 ± 11.04 %; pD2 = 6.62 ± 0.19; n = 5) when compared to the CTL group (Emax = 113.24 ± 5.59 %; pD2 = 7.72 ± 0.16; n = 4) (Figs. 3G and 3H). Furthermore, considering this result, we have added a topic (see sentence 419-423) that discusses the relationship between the loss of erectile components and the relaxing response to SPN.

“This reduction was able to alter the functionality of the smooth muscle of the corpus cavernosum since there was a significant reduction in the potency of the SPN in the DGAL group when compared to the CTL. Nevertheless, most likely, the morphological changes are not at a level to create functional changes sufficient to alter the maximum response to SNP.”

---

## [Editor Report · Decision Letter 1]

9 Feb 2021

PONE-D-20-33703R1

D-[+]-Galactose-induced aging:  a novel experimental model of erectile dysfunction

PLOS ONE

Dear Dr. Medeiros,

Thank you for submitting your manuscript to PLOS ONE. You did not answer to the comments of reviewer 1. Therefore, we invite you to submit a revised version of the manuscript that addresses the points raised by this reviewer:

The Authors have conducted a new experimental model of DE, using the D-(+)-Galactose aging model. The Authors found that the D-(+)-Galactose-induced aging represents a new experimental model suitable for the assessment of ED, evidenced by the hypercontractility of the corpora cavernosa, plus a significant endothelial dysfunction.

The logic and presentation of the key idea is not bad. I have following concerns about this work at least in its present form.

1. There was a significant difference in body weight between the CTL group and the DGAL group.

2. There are some typing, writing and grammatical errors in this manuscript. This work does require an extensive edit by a native-English speaker.

We look forward to receiving your revised manuscript.

Kind regards,

Michael Bader

Academic Editor

PLOS ONE

---

## [Author Response · Author response to Decision Letter 1]

16 Mar 2021

Response to the Reviewer 1

We thank reviewer 1 for the thoughtful suggestions and for the good insights for improving our manuscript. Here we tried to answer in the best form possible to all the concerns.

1. There was a significant difference in body weight between the CTL group and the DGAL group.

Answer: According to the tables below, the animals both in the CTL and DGAL groups presented similar graduated increases in their body weights without statistical differences (n = 5; p > 0.05) (Fig 1C) (see Results Section, Sentence 206-207). 

Statistical analyzes were performed using the bidirectional analysis of variance followed by Bonferroni´s test for post hoc comparisons, using the GraphPad Prism 7.0 program. Differences between groups were considered statistically significant at P <0.05.

Time (week) CTL GROUP DGAL GROUP Statistics

 Mean (g) SEM N Mean (g) SEM N Significant? P Value

 1 272,00 6,89 5 251,00 5,74 5 No >0,9999

 2 284,00 9,13 5 272,60 7,81 5 No >0,9999

 3 297,60 13,27 5 282,20 12,96 5 No >0,9999

 4 317,80 15,10 5 293,00 13,23 5 No >0,9999

 5 328,20 16,56 5 295,80 12,59 5 No 0,3342

 6 335,60 16,98 5 313,00 14,12 5 No >0,9999

 7 344,80 17,34 5 327,80 12,72 5 No >0,9999

 8 359,00 14,43 5 341,60 11,16 5 No >0,9999

2. There are some typing, writing and grammatical errors in this manuscript. This work does require an extensive edit by a native-English speaker.

Answer. The manuscript underwent an extensive review by a native speaker. A declaration of manuscript translation-corrections was included as a second supporting information file.

---

## [Editor Report · Decision Letter 2]

19 Mar 2021

D-[+]-Galactose-induced aging:  a novel experimental model of erectile dysfunction

PONE-D-20-33703R2

Dear Dr. Medeiros,

We’re pleased to inform you that your manuscript has been judged scientifically suitable for publication and will be formally accepted for publication once it meets all outstanding technical requirements.

Kind regards,

Michael Bader

Academic Editor

PLOS ONE
---

## [Editor Report · Acceptance letter]

29 Mar 2021

PONE-D-20-33703R2 

D-(+)-Galactose-induced aging: a novel experimental model of erectile dysfunction 

Dear Dr. de Medeiros:

I'm pleased to inform you that your manuscript has been deemed suitable for publication in PLOS ONE. Congratulations! Your manuscript is now with our production department. 

Kind regards, 

on behalf of

Prof. Michael Bader 

Academic Editor

PLOS ONE